# MiRNA-Mediated Regulation of S100B: A Review

**DOI:** 10.3390/neurosci6030075

**Published:** 2025-08-08

**Authors:** Animesh Dali, Suhana Basnyat, Rachel Delancey, Nipun Chopra

**Affiliations:** 1Department of Biochemistry, DePauw University, Greencastle, IN 46135, USA; 2Immunology Graduate Program, University of Cincinnati College of Medicine, Cincinnati, OH 45267, USA; 3Department of Molecular Physiology and Biological Physics, University of Virginia, Charlottesville, VA 22904, USA; 4Department of Psychology and Neuroscience, DePauw University, Greencastle, IN 46135, USA

**Keywords:** MiRNA, microRNA, S100β, inhibition, mRNA transcription, downregulation

## Abstract

S100β is a significant signaling molecule and biomarker that is primarily expressed in the brain. At low physiological concentrations, S100β induces astrocyte maturation, microglial migration, and neural proliferation. However, high concentrations activate inflammatory and pro-apoptotic pathways. Due to this dual role, increased research is being invested into the role of S100β in neuronal homeostasis and inflammation. In fact, increased S100β expression is seen in many neuropathologies including Alzheimer’s disease, Parkinson’s disease, cerebral ischemia, and traumatic brain injury. High S100β is generally associated with worsened disease outcome. Here, we provide an overview of the structure and role of S100β in various pathways, particularly in the context of neurological disorders. Modulation of S100β levels also holds promise as a therapeutic strategy. Micro-RNAs (miRNA) post-transcriptionally regulate gene expression and provide a novel approach reduce excess S100β protein. However, much of this research is still in its infancy. We outline current studies identifying miRNA in human and animal models of various neurological disorders. S100β itself has several predicted miRNA interactions although most have not yet been directly validated. Together, we compile the literature identifying S100β and miRNAs to guide future research in this field. We also comment on the feasibility and future uses of miRNA for pharmaceutical regulation of S100β, particularly for neurological treatments.

## 1. Introduction

### 1.1. The S100β Protein

S100 calcium-binding protein B (S100β) is a member of the highly conserved Calcium (Ca^++^) and Zinc (Zn^++^) binding S100 family of proteins and plays a vital role in cell differentiation, proliferation, and apoptosis [1,2,3]. It is expressed in the cytoplasm and nucleus of a wide range of cell lines and tissues including neural cells and astrocytes [4]. The secretion of S100β is triggered by intracellular calcium mobilization from the endoplasmic reticulum (ER) and is modulated by metabolic stress and cell injury [5,6,7,8,9,10]. Notably, S100B lacks a classical signal peptide, supporting its export through noncanonical, ER-independent, secretory mechanisms, such as exosome-mediated release or passive release across the plasma membrane, depending on the health of the secreting cell [6,11,12,13].

The S100β protein consists of 2 identical chains, each made up of 92 amino acid polypeptides that include four ɑ-helices and two β-sheets held together by a central hinge region (Figure 1) [14,15,16]. The two β-sheets are between helices 1 and 2, and 3 and 4, with a linker region between the two groups, resulting in two helix-loop-helix EF-hand motifs which allow for reorientation after binding to calcium, allowing the S100β to bind to target proteins including tumor protein p53, capping protein Z (CapZ), receptor for advanced glycation end product (RAGE), nuclear dbf2-related (NDR) kinases, neurotensin, cathepsin L inhibitor, human homolog of murine double minute 2 (Hdm2), Hdm4, protein kinase Cα (PKCα), rod outer segment membrane guanylate cyclase type 1 (ROS-GC1), microtubule-associated protein tau (tau), melittin, amyloid-β (Aβ), interleukin (IL)-11, the serotonin 5-HT7 receptor, the dopamine D2 receptor, ribosomal protein S6 kinase α-1 (RSK1), voltage gated ether à go-go 1 (EAG1) potassium channels, and Ras GTPase-activating-like protein IQGAP1 (IQGAP1) [17]. The ability of S100β to interact with various peptide sequences in its vast array of biologically important target proteins enhances its highly pro-apoptotic properties [17,18].

As a calcium sensor protein, S100β undergoes conformational changes upon C binding to its EF-hand motifs, exposing hydrophobic surfaces that enable interactions with a wide range of target proteins [19]. These interactions regulate diverse intracellular processes, including cytoskeletal organization, enzyme activity, and cell migration. Notably, S100β binds and inhibits EAG1 potassium channels and interacts with proteins such as Ras GTPase-activating-like protein IQGAP1 and p53 in a Ca^++^ dependent manner, modulating cell polarity and suppressing tumor activity [20]. Another key role of Ca^++^ in S100B function is to promote its multimerization in high Ca^++^ and non-reducing conditions, which facilitates RAGE dimerization and activation, initiating downstream signaling pathways [4]. Zn^++^ binding further modulates S100β function by increasing its affinity for calcium, stabilizing its active structure to enable more efficient calcium sensing at physiological concentrations. As a result, zinc-bound S100β may play a role in regulating excitotoxicity due to changes in calcium concentrations [21].

**Figure 1 neurosci-06-00075-f001:**
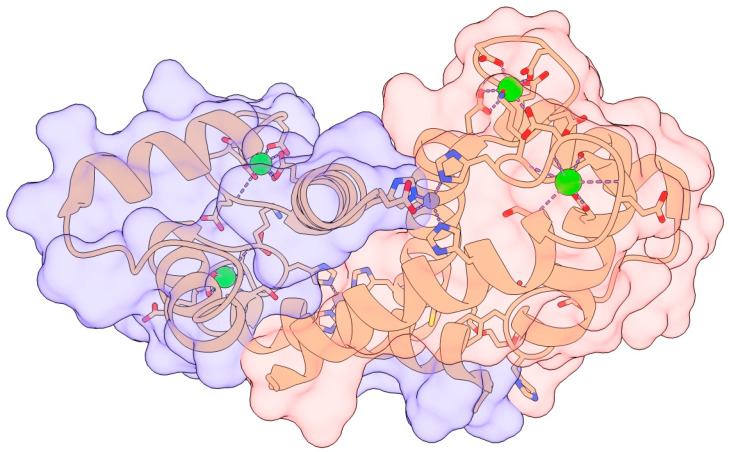
Two monomers dimerize to form the S100β protein. Four EF-hand motifs binding to Calcium (green) and Zinc (purple) binding regions of the protein can be observed. Figure produced using Chimera X Release 1.4. Created in BioRender. Chopra, N. (2025) https://BioRender.com/a05q149 (accessed on 30 June 2025) [22].

S100β has homologs in diverse species, including chimpanzee, Rhesus monkey, dog, cow, mouse, rat, chicken, and zebrafish. Typically, S100β in these species comprises 92 amino acids however exceptions exist in the dog and zebrafish. Canine S100β protein is composed of 105 amino acids, with additional residues occurring before the calcium-binding motif, while zebrafish S100β consists of 95 amino acids, featuring three extra residues after the calcium-binding motif. Across all conserved species, S100β plays a pivotal role in RAGE receptor binding, S100 protein binding, and calcium binding (GenBank) [23]. S100β mRNA has three distinct splice variants: S100β-201 (ENST00000291700.9), S100β-202 (ENST00000367071.4), and S100β-203 (ENST00000397648.1), characterized by transcript lengths of 1109, 971, and 839 nucleotides, respectively. The corresponding 3′ UTR lengths are 745, 489, and 82 nucleotides. Notably, Variants 201 and 203 share identical protein sequences, constituting the “normal” 92-residue chain, while Variant 202 diverges at position 47, featuring a distinct amino acid sequence and a total of 94 residues [24]. The half-life of S100β mRNA may be as long as 24 h [25], indicating a potential “stationary target” for mRNA-targeting medications.

### 1.2. S100β Is a Promising Therapeutic Target

The extracellular release of the S100β protein is controlled primarily by two conditions—metabolic stress and stimulation by external factors. When cells undergo metabolic stress, such as hypoxia or glucose deprivation during the developmental stages of astrocytes, S100β is released extracellularly. Similarly, extracellular S100β release occurs due to external stimulus such as serotonin (5 HT), glutamate, proinflammatory cytokines ILβ and tumor necrosis factor (TNF)-α, Aβ, lysophosphatidic acid, epicatechin, resveratrol, and increasing intracellular neuronal calcium concentrations [26].

RAGE is a membrane-integral protein primarily associated with activation of proinflammatory genes and is a receptor for S100β at a micromolar concentration [27]. Interaction of S100β and RAGE starts a cascade that results in production of high levels of free oxygen radicals, commonly known as reactive oxygen species (ROS). The activation of RAGE increases production of ROS catalysts such as inducible nitric oxide synthase (iNOS), an enzyme responsible for production of nitric oxide (NO) [3]. ROS such as NO are important neuronal cell signaling molecules and in high concentrations, can lead to the increased expression of proinflammatory factors such as the mitochondrial protein cytochrome c oxidase subunit II (CoxII) [28]. Upregulation of kinases such as CoxII increases ROS production in neurons, eventually leading to cell cycle arrest and neuronal death [29]. High concentrations of ROS induce increased mutation in mitochondrial DNA, resulting in mitochondrial dysfunction as changes in membrane permeability to calcium causes imbalance of Ca^++^ in the mitochondrial body. Mitochondrial dysfunction eventually leads to neuronal apoptosis, commonly seen in neurodegenerative diseases such as Alzheimer’s disease (AD), Parkinson’s disease (PD), and amyotrophic lateral sclerosis (ALS) [30].

The inflammatory cascades following oxidative stress caused by the upregulation of S100β can also disrupt intracellular Ca^++^ release by mobilizing calcium from intracellular inositol 1,4,5-trisphosphate-sensitive stores [31,32,33,34]. Disruption of Ca^++^ homeostasis has been linked to upregulation of Aβ and increased production of ROS (specifically NO) species [3,33]. Aβ induces apoptosis in human cells, microglial cells, and mouse models. Interestingly, research has shown that the S100β protein can bind an Aβ isoform, Aβ42, in their monomeric, oligomeric and fibril states, hindering their aggregation in low Aβ42 concentrations [35]. However, at high Aβ concentrations, there is evidence of the activation of the focal adhesion kinase (FAK), which in turn has been linked to the activation of the mitogen-activated protein kinase 3 (ERK1)/mitogen-activated protein kinase 1 (ERK2) pathway. The activation of the upstream ERK1/ERK2 results in the upregulation of nuclear factor κB (NF-κB). This increased expression of NF-κB is followed by its nuclear localization, which upregulates proinflammatory kinases such as TNF-α and IL-1β in glial cells (microglia and astrocytes) [29,34]. When C2C12 myoblast cells were treated with, Sulfasalazine (SSN), a NF-κB inhibitor, S100β mRNA and protein levels were downregulated, suggesting that NF-κB activity positively regulates S100β [36].

S100β also functions extracellularly, where changes in Ca^++^ and K^+^ levels can trigger its release from astrocytes. It facilitates astrocyte and microglia migration and can either exert a neurotrophic or neurotoxic function depending on its own extracellular concentrations [4]. At normal levels S100β promotes chemotaxis and quiescence of microglia, participates in differentiation and maturation of oligodendrocytes, and enhances astrocytic proliferation [35,37,38]. The inflammatory role of S100β is particularly important for microglial activation and polarization. As the resident immune cells of the brain, microglia undergo dynamic shifts between M1 and M2 states in response to external stimuli. The M1 phenotype is activated by signals from interferon gamma (IFN-y), TNF-a, and LPS to lead to a pro-inflammatory state. These microglia help initiate an immune response in response to cell damage. M2 microglia are associated with the release of anti-inflammatory and pro-survival cytokines in response to interleukin-4 (IL-4) and interleukin-13 (IL-13) [39]. Microglia in the M2 state help repair tissue by clearing cellular debris and modulating neuroinflammation. Both M1 and M2 microglia are important for an effective immune response and dysregulation of the balance between these states have been implied in several neuropathologies. In Alzheimer’s and Parkinson’s disease, M2 microglia attenuate damage by clearing misfolded proteins. M2 microglia also promote oligodendrocyte remyelination in multiple sclerosis. However, the accumulation of misfolded or aggregated proteins promotes a sustained pro-inflammatory M1 microglial state, which worsens inflammation and contributes to neuronal cell death [40]. As a result, M1 and M2 microglia phenotypes are largely associated with disease severity.

Several pathways such as NF-κB, JNK, and high ROS are important for both the M1 phenotype and S100β. Indeed, treatment of cultured microglia with S100β was sufficient to activate microglia and upregulate M1 gene expression (TNF-a, IL-6, iNOS), but downregulate M2 gene expression (IL-10, TGFβ). In a cerebral ischemia mouse model, injection of S100β significantly increased infarct size, percentage of TUNEL-positive neurons, and M1 microglial markers. Concomitantly, the inhibition of S100β reduced infarct size, percentage of TUNEL-positive neurons, and increased M2 microglial markers [41]. Brain tissue samples from patients with Parkinson’s disease show increased S100β protein levels. Knockout of S100β in mice partially rescued neuronal death in an MPTP Parkinson’s model [42]. Modulating S100β levels may help shift the balance between M1 and M2 microglial states, offering a potential therapeutic strategy after brain injury or ischemia.

S100β can bind to astrocyte-specific characteristic intermediate filament (IF) and glial fibrillary acidic proteins (GFAP), which control brain homeostasis. GFAP S100β oligomerization affects polymerization of tubulins and DNA repair [12]. This dysfunction of microtubules can lead to synaptic impairment and cell death [43]. Dysregulation of S100β can lead to hyperphosphorylation of tau, which destabilizes assembly of microtubules [44]. In diseases such as AD, hyperphosphorylation of tau (p-tau) results in formation of tau tangles that accumulate in neurons [45]. In cultured human neuronal cells, micromolar concentrations of S100β can essentially “hijack” canonical Wnt signaling. By stimulating c-Jun N-terminal kinase (JNK) phosphorylation, which increases expression of Dickopff-1 (DKK-1), S100β can increase the phosphorylation of glycogen synthase kinase 3β (GSK3β). GSK3β is canonically known for degrading β-catenin and is the main target in the Wnt pathway. However, it is also the primary kinase to phosphorylate tau [46]. As a result, downregulation of S100β can be an important target to better control hyperphosphorylation of the tau protein and subsequent formation of tau tangles.

S100β and NF-κB in myoblast cells also interact to increase NF-κB activity. An increase in the MKP-1-p38 MAPK cascade downregulates S100β mRNA, forming a closed loop cascade with S100β and normal functioning of NF-κB. However, in high concentrations of S100β, nuclear localization of NF-κB increases, resulting in its increased transcriptional activity [36]. Increased NF-κB transcriptional activity results in upregulation of proinflammatory kinases such as CoxII, TNF-α, and IL-1β, eventually leading to the upregulation of S100β [29,34]. S100β can also control myoblast proliferation and differentiation by binding basic fibroblast growth factor (bFGF). The S100β-bFGF complex then recruits basic fibroblast growth factor receptor 1 (FGFR1). Activation of the FGFR1 increases myoblastic cell mitogenesis. The S100β in this complex is also able to bind to RAGE to form an S100β-bFGF-FGFR1-RAGE complex, which results in the inhibition of the RAGE protein myoblastic differentiation activity [47].

S100β also disrupts p53 function through multiple mechanisms, including direct binding to its C-terminal oligomerization domain, impeding both oligomerization and the subsequent activation of p53, while also hindering its activation, phosphorylation, and regulation mediated by protein kinase C [48,49,50,51]. This mechanism, however, has not been deeply investigated in neuronal tissue. Although this is contrary to earlier stated roles in inducing apoptosis, this mechanism could further elucidate a role of S100β in cancer biology. Indeed, one study found that s100β-p53 complexes are anti-apoptotic, increasing survival for melanoma cells [49].

In short, S100β in neuronal tissue is mainly a pro-apoptotic protein and is upregulated in neurodegenerative diseases and injuries such as traumatic brain injury (TBI), AD, Down syndrome, ALS, multiple sclerosis, schizophrenia, major depressive disorder, cerebral stroke, Parkinson’s disease, IS, cartilage injury and hypoxic-ischemic encephalopathy (HIE) [3,46,48,52]. Targeted degradation of S100β, particularly through microRNA-mediated repression of its mRNA, represents a promising strategy for mitigating neurodegenerative pathology by curbing its pro-apoptotic signaling cascade (Figure 2).

### 1.3. MicroRNA Regulation of Protein Expression

MicroRNAs (miRNAs, miR-) are small endogenous RNA molecules, typically around 19–25 nucleotides in length, which play a pivotal role in modulating protein expression [50,51,53,54,55]. These molecules typically provide target specificity for the RNA-induced silencing complex (RISC). MiRNAs are synthesized through two pathways: the canonical and non-canonical pathway. In the canonical pathway, RNA polymerase II (POLII) initiates transcription of primary miRNA (pri-miRNA), which is cleaved and processed by several enzymes, mainly DROSHA, to produce a duplex [50]. The miRNA duplex interacts with RISC, where it forms a complex with argonaute 2 (AGO2). AGO2 destabilizes the miRNA duplex [56]. Following this, RNA helicase unwinds the destabilized miRNA duplex, and an endoribonuclease complex comprising translin-associated factor X (TRAX), TRANSLIN, and heat shock protein 90 (HSP90) degrade the passenger strand [50]. In the non-canonical pathway, pre-miRNAs are processed from very short introns called mirtrons by spliceosome and then debranched by the lariat debranching enzyme, inherently skipping DROSHA cleavage [50]. The pre-miRNA then gets exported to the cytoplasm by XPO5 and is then processed similarly to the canonical pathway.

Binding sites of miRNA on the target mRNA 3′-UTR or 5′-UTR are characterized by complementarity to the miRNA. The three canonical binding sites are the 8mer, 7mer-m8, and 7mer-A1 binding sites. The marginal sites, 6mer and offset 6mer have a 6-nucleotide match to the miRNA seed region. The offset 6mer is shifted by one nucleotide to the seed region. These sites are weaker than the canonical binding sites [51,57,58] (Figure 3). The AGO-miRNA complex positions nucleotides 2–8 for pairing with mRNAs, constituting the 8mer binding site. Nucleotide 1 is turned away, not available for pairing, while nucleotides 9–11 face away from incoming mRNAs [51]. The 8mer site for binding is recognized by the binding pocket and can either form a helical segment with mRNAs at positions 13–16 without disturbing the remaining miRNA or induce a conformational change, allowing the miRNA and mRNA to wrap around each other. The AGO protein locks down the perfectly paired miRNA-mRNA duplex, positioning it in its active site for cleavage performed by either AGO itself or another protein recruited by the AGO-miRNA complex [51]. Under most circumstances, imperfect base pairing with target mRNA results in translational repression, but extensive complementarity can instead degrade the miRNA [59]. While most miRNA activity represses protein expression [60,61], in some cases miRNA elevates protein expression [62]. It is this potential for controlling protein levels that is currently under exploration in other fields, such as oncology, for the use of miRNA as a disease treatment [63].

## 2. The miRNA-S100β Axis in Neurological Disease

S100β is a biomarker for disability and mortality-causing cases such as ischemic shock (IS), cartilage injury, and HIE and is a recommended target for inhibition to alleviate neuronal death [3]. In a study conducted using samples from IS patients immediately following admission, serum S100β levels were upregulated, while serum miR-602 levels were downregulated shortly after injury. After 3 months, serum miR-602 and S100β serum levels did not show significant change compared to non-IS subjects [46]. The predictive model TargetScan predicts miR-602 to have a 7mer-A1 binding site at the 3′ UTR of S100β-201 and S100β-202 [57]. miR-602 reduces ROS levels in oxygen and glucose deprivation/reoxygenation (OGD/R) models by increasing transcription activity ofNRF2/ARE (nuclear factor erythroid 2-related factor 2) and in turn, NRF2 expression [64]. In the nucleus, NRF2 binds to electrophile responsive elements (EpRE), forming a heterodimer with small Maf proteins, which results in the transcription of the antioxidant gene heme oxygenase 1 (HO-1) among other detoxifying genes [65,66]. HO-1 alleviates damage caused by oxidative stress through downregulation of inflammatory cytokines IL-6 and TNF-α [67].

miR-302a-3p is downregulated in IS mouse models. In middle cerebral artery occlusion (MCAO) mouse models, upregulation of miR-302a-3p was consistent with decreased concentration of S100β, GFAP, and the proinflammatory proteins TNF-α/IL-1β. Details of the mechanism through which S100β is downregulated by miR-302a-3p remain unknown. This reduction in protein concentration was associated with an increase in recovery from CIS-induced brain injury [68].

In patients with craniocerebral injury (CI), serum levels of miRNA-124, miRNA-210, Janus kinase 2 (JAK2), signal transducer and activator of transcription 3 (STAT3), mitogen-activated protein kinase kinases (MEK), ubiquitin carboxy-terminal hydrolase L1 (UCH-L1), GFAP, S100β, Tau and ERK1/2 were all higher in CI group compared to the healthy controls [52]. This finding suggests that the JAK2-STAT3 cascade and MEK-ERK1/2 cascade is activated following CI. These cascades upregulate the expression of inflammatory cytokines IL-1β, IL-6, and TNF-α [69,70]. The miRNAs miR-124 and miR-210 were overexpressed following the upregulation of these inflammatory cytokines following CI. As a result, miR-124 and miR-210 could be potential biomarkers for CI [52]. Since S100β is positively correlated with these miRNAs, is also worth investigating the interaction between miR-124/miR-210 targets and S100β.

miRNA-328 and miRNA-378 are linked with brain metastases (BM) in non-small-cell lung cancer (NSCLC) [71,72]. Upregulation of miR-328 in A549 cells resulted in increased PKCα concentrations [71]. Similarly, vascular endothelial growth factor (VEGF) 3′-UTR binds to miR-378, resulting in upregulated VEGF expression [71]. This overexpression of PKCα was followed by gene dysregulation causing irregular function of the IL-1 and VEGF signaling pathway, which resulted in loss of cell adhesion and increased cell migration [71,72]. Further investigation into this pathway has identified ser/thr protein kinase Raf-1 and the anti-apoptotic protein B-cell lymphoma 2 (Bcl-2) as potential targets of PKCα [73]. Phosphorylation of Raf-1 during apoptotic events could result in the inactivation of the pro-apoptotic mitochondrial membrane protein BAD which could promote cell viability. Bcl-2 is co-localized with PKCα in the mitochondrial membrane and is predicted to be able to phosphorylate Bcl-2 in the Ser 70 residue and increase its activity [73,74]. It turns out that elevated Bcl-2 protein levels associate with resistance to S100β-mediated cell death in human neuronal cell line NT2/D1 [75]. Downregulation in PKCα has also been linked with the increased activity of the p53, which could result in increased p53-dependent apoptosis [73].

### MiRNAs That Directly Target S100β mRNA

As we have seen, S100β is upregulated in neurodegenerative diseases and CNS injuries. Downregulation of the S100β protein can potentially mitigate harmful effects associated with its overexpression. An emerging method of downregulating S100β is to explore miRNA regulation of S100β mRNA translation. Probing TargetScan 8.0 with the human S100β sequence predicted interaction with one well-conserved miRNA species, miR-330-3p [57,58]. However, such conservation merely summarizes how widespread a given predicted binding may be across species and does not necessarily indicate potential medical utility, particularly if this utility might pertain to a condition not found in non-human species (e.g., Alzheimer’s disease). If we compare the pertinent scores of predicted interactions to those for miR-330-3p (Table 1 and Table A1), we find that 45 other miRNA species are predicted to have at least as low a weighted context^++^ score (indicating predicted strength of target repression) and at least as high a context^++^ percentile (indicating favorability of the context^++^ score). Sites are distributed throughout the S100β 3′-UTR (Figure 4).

MiR-330-3p levels decreased significantly in cartilage and synovial tissues of rabbit models who had cartilage injury compared to mock treated animals. Introducing an miR-330-3p overexpression vector to cartilage injury animals reduced S100β concentrations [76]. Inhibiting miR-330-3p with long noncoding RNA (lncRNA) X-inactive specific transcript (XIST) resulted in increased levels of S100β in *e* cardiomyocyte mice models, suggesting lncRNA such as XIST can imitate miRNA sponges to prevent target mRNA binding [77].

S100β concentrations are upregulated in HIE. In newborn HIE patients, serum miR-199a concentrations decreased significantly in moderate and severe cases of HIE. In the same subjects, serum S100β and neuron-specific enolase (NSE) levels were upregulated in moderate and severe cases. Taken together, the upregulated S100β and NSE levels alongside downregulated miR-199a levels gave high diagnostic accuracy in HIE newborns, suggesting that miR-199a might be interacting with S100β as a potential inhibitor [48]. miR-199a-5p, may play a protective role in cerebral ischemic injury by inhibiting the expression of DDR1, a tyrosine kinase receptor which increases the expression of proinflammatory kinases such as TNF-a, IL-6, and IL-1β in mice models [81]. In a hypoxia/ischemia induced cerebral palsy rat model, miR-135b downregulated s100β and promoted differentiation in neural stem cells [83].

In amyotrophic lateral sclerosis (ALS), a specific mutation of superoxide dismutase 1 (SOD1) plays an important role, and downregulating mutant SOD1 (mSOD1) can slow ALS progression through reduced motor neuron (MN) damage and loss [84,85,86]. In ALS, neuroinflammation and glial cell activation contribute to MN degeneration [87]. mSOD1 increased expression of NF-κB, pro-inflammatory factors including IL-1β, TNFα, miRNA species miR-155, miR-125b, miR-146a, miR-21, miR-124, and S100β in the symptomatic phase [79]. mSOD1-induced mouse models had an increase in NF-κB, HMGB1, connexin 43, and S100β expression in the symptomatic stage, while displaying MN damage and loss coupled with decreased expression of synaptic protein and miRNA species miR-125b, miR-21, miR-146a, and of GFAP, and glutamate transporters. Furthermore, astrocytes in the cortex undergo a neurotoxic transformation in the early stages of ALS. Loss of miR-146a could be a contributory factor in the transformation of the astrocytes and this miRNA could be a potential therapeutic target [80]. Transfection with pre-miR-146a and treatment with dipeptidyl vinyl sulfone (VS) of mSOD1 mice astrocyte models resulted in upregulation of GFAP and restoration of normal neuronal activity. Upregulating miR-146a reduced S100β, connexin 43, TNF-α, and iNOS mRNAs [88]. Anti-miR-124 treatment of mSOD1 mice MN found reversed MN degeneration and glial cell activation. TREM2 and IL-10 genes were upregulated and GFAP, Connexin 43, S100β protein levels and inflammatory miRNAs miR-146a, miR-155, and miR-21 were reduced [89].

A little studied small RNA species is circular RNA (circRNA), in which the mature molecule’s 3′ and 5′ ends are covalently linked [90]. One example, circ_MUC16 may regulate miR-1182. Specifically, reduction of Circ_MUC16 increased miR-1182 expression and upregulation of circ_MUC16 decreased miR-1182 expression. miR-1182 regulates S100β through a specific binding site at the S100β 3′ UTR [78]. In short, there are multiple examples of small RNA species regulating S100β levels in connection with multiple disorders, some of which may prove useful therapeutic avenues.

## 3. Discussion

Elevated S100β expression is a potential biomarker for neurotrauma and has been linked with increased proapoptotic activity. There is growing interest in therapeutic strategies aimed at reducing S100β concentrations. One emerging therapeutic approach involves modulating the S100B–RAGE axis, particularly in the context of traumatic brain injury. RAGE activation after TBI contributes to neurovascular unit damage, blood–brain barrier disruption, and neuroinflammation [91,92,93,94,95]. Inhibiting RAGE in animal models, via gene knockouts genetically or with antagonists like FPS-ZM and papaverine, reduces blood–brain barrier damage, brain edema, and neuroinflammation. In cellular models, RAGE is upregulated in brain tissue after TBI, especially in microglia and pericytes near the injury site. Blocking RAGE reduces the number of activated glial cells and apoptotic cells, while increasing neuron survival [91,93,96]. Multiple miRNAs, including miR-214, miR-185-5p, miR-107, miR-21, miR-155, and miR-181a, are known to modulate RAGE or its signaling pathways in cancer biology, while their role in neuroinflammation or TBI remains unknown [97,98,99,100,101].

Here, we primarily focus on inhibition of S100β mRNA translation through the employment of S100β-targeting miRNAs due to three factors: (1) the complexity of RAGE-dependent signaling makes it a tricky therapeutic target [94,102], (2) S100B levels are greatly elevated after a TBI, and (3) S100B is neurotrophic at lower concentrations and neurotoxic depending at higher concentrations, thereby modulation of this protein would be a good therapeutic goal. It is not uncommon for multiple miRNAs to simultaneously target the same mRNA and multiple mRNAs to be targeted by the same miRNA—a dynamic and multifaceted system of interactions between miRNA and their mRNA targets. This concurrent targeting introduces a level of competition among different miRNA species, which, when applied in clinical models outside of a controlled environment, can yield issues regarding efficiency and specificity. On the other hand, this could also provide opportunities for both efficiency and specificity, wherein a miRNA cocktail could be formulated where multiple sites could be utilized in synergy.

That being said, therapeutic miRNA regulation of S100β is in its nascent stages, multiple clinical hurdles still exist. These include the promiscuity of miRNA targeting and miRNA delivery across the blood–brain barrier. Although much progress in circumventing the blood–brain barrier has been led by brain tumor research [103], as of this paper’s composition, no phase 3 or 4 studies have been registered at any time with the FDA that use miRNA as an intervention (National Library of Medicine) [104]. Of the Phase 1 and 2 studies, all are pharmacokinetic or pharmacodynamic in nature and do not address efficacy as a primary outcome (National Library of Medicine) [104].

There exists a notable gap in experimental elucidation of miRNA interaction with S100β mRNA. A reliance on computational predictions in numerous studies magnifies the risk of false positives and non-specific interactions. In the predominant transcript of S100β mRNA (S100β-201), only a single miRNA, miR-330-3p, has been exhaustively confirmed in work that dissects the specific miRNA target site on the S100β 3′-UTR. Other experimental work we recounted that explored miRNA interventions on S100β levels has been primarily associative, without explicitly confirming molecular site interactions. The absence of experimental validation for established miRNA targets emphasizes the critical necessity for conducting studies confirming specific miRNA species binding and degrading S100β mRNA.

An issue also arises when it comes to S100β expression itself. Each of the 3 mRNA isoforms have a different UTR. While this is useful to native cellular control of stability of S100β expression, it poses an additional challenge when considering miRNA treatments. There is little characterization of which transcripts are expressed in which cells under which conditions. As a result, we do not know which UTR sequence to target with a treatment, such as neurons undergoing apoptosis vs. metastasizing lung cancer. To successfully utilize miRNA therapies, full characterization of S100β expression is required.

## 4. Conclusions and Future Directions

In conclusion, while targeting S100β via miRNA-based therapies presents a compelling strategy to mitigate neuroinflammation and pro apoptotic signaling following traumatic brain injury, substantial gaps remain in regulating miRNA-mRNA interactions as a result of overlapping targeting, limited UTR characterization across transcript variants, and scarce experimental validation. Moreover, clinical hurdles such as delivery across the blood–brain barrier and regulatory progression beyond early phase trials underscores the need for further research. While we focused on microRNA that modulate S100B in this review paper, other work should also independently focus on the potential RNA modulation of the RAGE receptor and downstream effectors.

A final consideration is that miRNA-target interactions do not exist in “pure” systems. Every cell contains a vast mixture of potential mRNA targets for any given miRNA, and any given miRNA will target a mixture of mRNAs. In addition to this theoretical truism is the simple fact that levels of mRNAs and miRNAs differ not only from tissue to tissue but can be perturbed in clinical conditions. A high background level of a miRNA could create an impractical hurdle to develop an effective supplement, and the non-linear kinetics of miRNA-mRNA interactions further complicate clinical titration [105]. However, these same complications could potentially be harnessed to tailor miRNA-based treatments to specific tissues and conditions. In any case, our current lack of knowledge, both on the specifics if miRNA-S100β interactions and the general details of miRNA therapies calls for further work.

## Figures and Tables

**Figure 2 neurosci-06-00075-f002:**
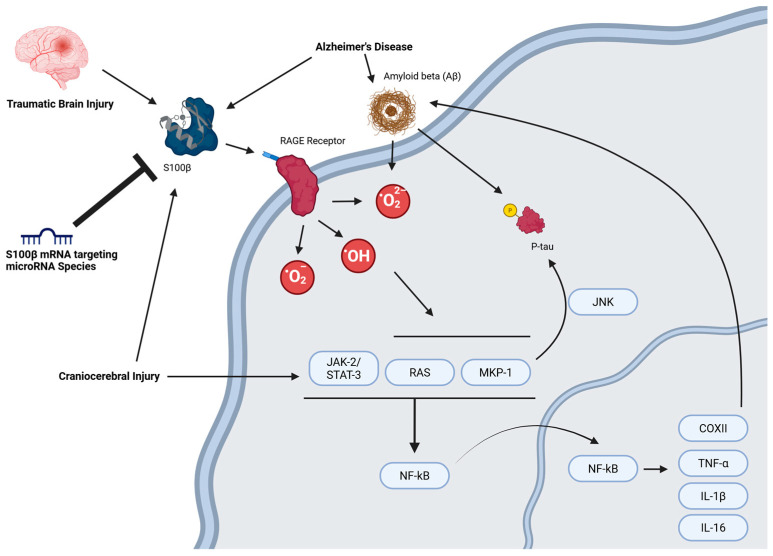
MicroRNA-mediated regulation of S100B-induced apoptotic signaling. Created in BioRender. Chopra, N. (2025) https://BioRender.com/pirn28g (accessed on 30 June 2025).

**Figure 3 neurosci-06-00075-f003:**
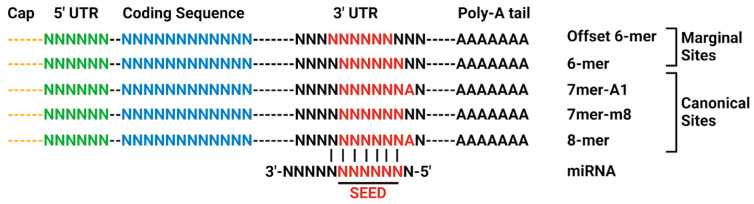
Canonical site types for miRNA-mRNA 3′-UTR interaction. Adapted from TargetScan. Created in BioRender. Chopra, N. (2025) https://BioRender.com/o98h678 (accessed on 30 June 2025).

**Figure 4 neurosci-06-00075-f004:**
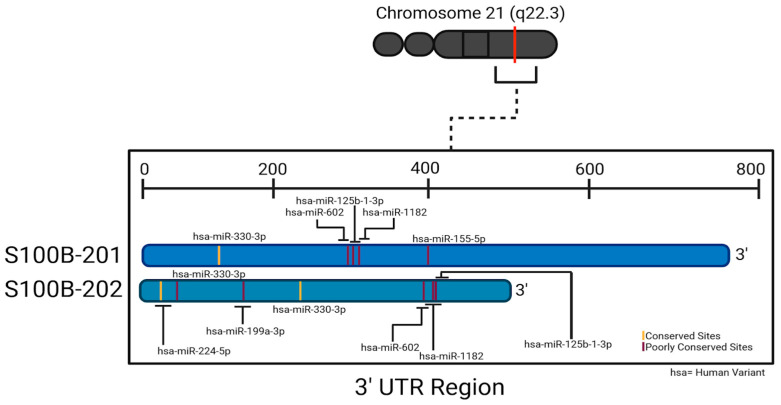
Two splice variants of human S100β 3′-UTR and respective binding sites to the conserved miRNA species and selected poorly conserved species. Created in BioRender. Chopra, N. (2025) https://BioRender.com/x73p821 (accessed on 30 June 2025).

**Table 1 neurosci-06-00075-t001:** Selected miRNA-S100B 3′ UTR interactions.

mRNA Species	miRNA Name	Binding Pattern *	3′ UTR Position	Seed Match	Site Type	Reference
S100B-201	hsa-miR-330-3p	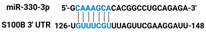	127-133	7mer-m8	Conserved	[57,76,77]
hsa-miR-1182	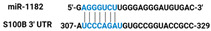	308-315	8mer	Poorly Conserved	[57,78]
hsa-miR-602	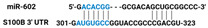	302-308	7mer-A1	Poorly Conserved	[46,57,64]
hsa-miR-125b-1-3p	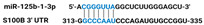	314-320	7mer-m8	Poorly Conserved	[57,79,80]
hsa-miR-155-5p	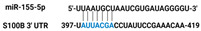	398-404	7mer-m8	Poorly Conserved	[57]
S100B-202	hsa-miR-330-3p	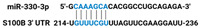	215-221	7mer-m8	Conserved	[56,75,76]
hsa-miR-1182	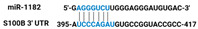	396-403	8mer	Poorly Conserved	[57,78]
hsa-miR-602	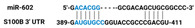	390-396	7mer-A1	Poorly Conserved	[46,57,64]
hsa-miR-125b-1-3p	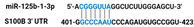	402-408	7mer-m8	Poorly Conserved	[57,79,80]
hsa-miR-224-5p	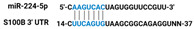	15-21	7mer-m8	Conserved	[57]
	hsa-miR-199a-3p	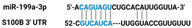	53-59	7mer-m8	Poorly Conserved	[48,57,81]

* Interactions predicted via TargetScan. Created in BioRender. Chopra, N. (2025) https://BioRender.com/u38d591 (accessed on 30 June 2025) [82].

## Data Availability

The original data presented in the study are openly available in https://clinicaltrials.gov/ and https://www.targetscan.org/ (accessed on 5 July 2025).

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
