# Peer review of "MiRNA-Mediated Regulation of S100B: A Review"

_neurosci, 2025, doi:10.3390/neurosci6030075_

Round 1
Reviewer 1 Report
Comments and Suggestions for Authors The manuscript by Nipun Chopra et al., entitled "MiRNA-Mediated Regulation of S100B: An Overview" describes the use of various MiRNA roles in neurodegenerative diseases. The manuscript is well written. Incorporating the minor changes mentioned below will further improve the manuscript.- I recommend including a table of selected miRNAs targeting S100B, along with their mechanisms of action and the systems in which they were tested.
- I recommend including a representative figure that illustrates the role of miRNA‑S100B in various neurodegenerative diseases and highlights the pathways involved.
- I recommend moving Table A1 "TargetScan predictions from S100B sequence" to the supplementary information rather than keeping it in the main manuscript.
Author Response
- I recommend including a table of selected miRNAs targeting S100B, along with their mechanisms of action and the systems in which they were tested.
RESPONSE: We thank the reviewer for this recommendation. Our updated Table 1 includes experimentally validated miRNA that target S100B. We also now provide the 3'UTR binding site (predicted or confirmed) to indicate mechanism of action. - I recommend including a representative figure that illustrates the role of miRNA‑S100B in various neurodegenerative diseases and highlights the pathways involved.
RESPONSE: We thank the reviewer for this idea. We have now added an additional figure (Figure 2) which ties the role of 3 neuroscience relevant diseases (we have chosen not to include other S100B-relevant disease states, such as cancer), and downstream signaling pathways relevant to S100B-RAGE receptor. - I recommend moving Table A1 "TargetScan predictions from S100B sequence" to the supplementary information rather than keeping it in the main manuscript
RESPONSE: Thank you for the suggestion. Table A1 is a part of the appendix.
Reviewer 2 Report
Comments and Suggestions for Authors
The manuscript “MiRNA-Mediated Regulation of S100B: An Overview” by Dali is a review article which provides an overview of the structure and role of S100β, particularly in the context of neurological disorders. The authors also summarize the extant literature that identifies miRNA regulating S100β levels, and subsequently comment on future uses of miRNA for pharmaceutical regulation of S100β. In general, this article is critical in this field and contains essential contents. I have minor concerns before this manuscript is accepted for publication.
The abstract is too short. The authors should include more detailed information.
Discussion, Conclusion and Future Directions are included in the same section. Can the authors describe the Discussion, Conclusion and Future Directions in the different sections?
Author Response
The abstract is too short. The authors should include more detailed information.
RESPONSE: We thank the reviewer for their attention to detail. The abstract has now been lengthened to better represent the article.
Discussion, Conclusion and Future Directions are included in the same section. Can the authors describe the Discussion, Conclusion and Future Directions in the different sections?
RESPONSE: We thank the reviewer for this suggestion. The Discussion has now been separated from the Conclusion and Future Directions section.
Reviewer 3 Report
Comments and Suggestions for Authors
The mini-review submitted for consideration by Animesh Dali and co-authors, raises the question about the correction of S100b levels, associated with the development of various CNS pathologies, through the effects of various miRNAs. The use of miRNAs in the therapy of neuropathologies is a relatively new, rapidly developing field, which determines the relevance of this topic. Overall, the work is quite interesting but requires some revisions.
- Although the authors indicate that S100b is a calcium-binding protein, as well as a protein capable of binding zinc ions, the authors do not describe the role of these ions in protein functioning and signaling functions. This information should be added to the text. Additionally, the phrase "also disrupt intracellular Ca++ release" requires clarification as to what specific release is meant.
- In the review, among the functions of S100b, its influence on microglial polarization is not mentioned, namely that changes in microglial phenotype occur in various neuropathologies, and modulation of this process can be a target for various types of therapy, including miRNA effects.
- The caption to Figure 1 contains the phrase "a functional S100β enzyme." As is known, S100b does not possess enzymatic activity, but if the authors have other information on this matter, it should be presented in the review.
- Since S100b, in addition to participating in the modulation of intracellular signaling cascades, can be secreted by cells, as noted by the authors, it would be good to add mechanisms of exocytosis. It would also not be superfluous to comment on the prospects of targeting not S100b itself through miRNA, but the RAGE receptor. S100b itself is considered as a diagnostic criterion, and its reduction, for example in serum, would not be an indicator of positive disease outcome, but simply a consequence of its lower release as a result of miRNA therapy. In this regard, targeting not S100b itself, but its targets, could be a more effective approach. The authors should discuss this in the text and justify the appropriateness of the approach they propose within the framework of the review.
Author Response
1. Although the authors indicate that S100b is a calcium-binding protein, as well as a protein capable of binding zinc ions, the authors do not describe the role of these ions in protein functioning and signaling functions. This information should be added to the text.
RESPONSE: Thank you to the reviewer for this helpful feedback. We have now added additional paragraph (see below and on Page 1) to accurately capture the role of Zn and Ca in the function of S100B protein.
S100B is a member of the highly conserved calcium (Ca++) and zinc (Zn++) binding S100 family of proteins and plays a vital role in cell differentiation, proliferation, and apoptosis. As a calcium sensor protein, S100B undergoes conformational changes upon Ca++ binding to its EF-hand motifs, exposing hydrophobic surfaces that enable interactions with a wide range of target proteins. These interactions regulate diverse intracellular processes, including cytoskeletal organization, enzyme activity, and cell migration. Notably, S100B binds to and inhibits EAG1 potassium channels and also interacts with proteins such as IQGAP1 and p53 in a Ca++ dependent manner, modulating cell polarity and suppressing tumor activity. S100B also functions extracellularly, where changes in Ca++ and K+ levels can trigger its release from astrocytes. A key role of Ca++ in S100B function is to promote its multimerization in high Ca++ and non-reducing conditions, which facilitates RAGE dimerization and activation, initiating downstream signaling pathways.
Zn++ binding further modulates S100B function by increasing its affinity for calcium, stabilizing its active structure to enable more efficient calcium sensing at physiological concentrations. By altering calcium signaling, zinc-bound S100B may play a role in regulating excitotoxicity. Thus, the crosstalk between calcium and zinc binding is essential for the activation and function of S100B in both intracellular and extracellular signaling.
2. Additionally, the phrase "also disrupt intracellular Ca++ release" requires clarification as to what specific release is meant.
RESPONSE: We thank the reviewer for identifying this missing context. We have now updated the language for clarity by changing the sentence to, "... by mobilizing calcium from intracellular inositol 1,4,5-trisphosphate-sensitive stores" and adding an additional reference related to IP3.
3. In the review, among the functions of S100b, its influence on microglial polarization is not mentioned, namely that changes in microglial phenotype occur in various neuropathologies, and modulation of this process can be a target for various types of therapy, including miRNA effects.
RESPONSE: We thank the reviewer for this helpful feedback. While we struggled to capture the compendium of information related to the role of microglia, we have now added two additional paragraphs (see below and on page 4). Additionally, our new Figure 2 captures some of these signaling related processes.
The inflammatory role of S100β is particularly important for microglial activation and polarization. As the resident immune cell of the brain, microglia undergo dynamic shifts between M1 and M2 states in response to external stimuli. The M1 phenotype is activated by signals from interferon gamma (IFN-y), TNF-a, and IPS to lead to a pro-inflammatory state. These microglia help initiate an immune response in response to cell damage. M2 microglia are associated with anti-inflammatory and pro-survival cytokines in response to interleukin-4 (IL-4) and interleukin-13 (IL-13). Microglia in the M2 state clear out cell debris and resolve damage. Both M1 and M2 microglia are important for an effective immune response and dysregulation of the balance between these states have been implied in several neuropathologies. In Alzheimer’s and Parkinson’s disease, M2 microglia attenuate damage by clearing misfolded proteins. M2 microglia also promote oligodendrocyte remyelination in multiple sclerosis. However, aggregated proteins in many neuropathies encourage a prolonged M microglia state, exacerbating inflammation and cell death in the disease. As a result, M1 and M2 microglia phenotypes are largely associated with disease severity.
Several pathways such as NF-κB, JNK, and high ROS are important for both the M1 phenotype and S100β. Indeed, treatment of cultured microglia with S100β was sufficient to activate microglia and upregulate M1 gene expression (TNF-a, IL-6, iNOS), but downregulate M2 gene expression (IL-10, TGFβ). In a cerebral ischemia mouse model, injection of S100β significantly increased infarct size, TUNEL-positive neurons, and M1 microglial markers. Inhibition of S100β reduced infarct size, TUNEL-positive neurons, and increased M2 microglial markers. Brain tissue samples from patients with Parkinson's disease show increased S100β protein levels. Knockout of S100β in mice partially rescued neuronal death in an MPTP Parkinson’s model. Modulating S100β levels could prove to be a beneficial mechanism of modulating the balance between M1 and M2 microglia populations in various neuropathies.
4. The caption to Figure 1 contains the phrase "a functional S100β enzyme." As is known, S100b does not possess enzymatic activity, but if the authors have other information on this matter, it should be presented in the review.
RESPONSE: We thank the reviewer for their attention to detail. We have now changed the figure legend to say "... S100B protein"
5. Since S100b, in addition to participating in the modulation of intracellular signaling cascades, can be secreted by cells, as noted by the authors, it would be good to add mechanisms of exocytosis.
RESPONSE: We thank the reviewer for this helpful feedback. We have now added an additional paragraph (see below and on page 1).
S100B secretion is triggered by intracellular calcium mobilization from the endoplasmic reticulum, and is modulated by metabolic stress and cell injury. Notably, S100B lacks a classical signal peptide, supporting its export through noncanonical, ER-independent, secretory mechanisms, such as exosome-mediated release or passive release across the plasma membrane, depending on the health of the secreting cell.
6. It would also not be superfluous to comment on the prospects of targeting not S100b itself through miRNA, but the RAGE receptor. S100b itself is considered as a diagnostic criterion, and its reduction, for example in serum, would not be an indicator of positive disease outcome, but simply a consequence of its lower release as a result of miRNA therapy.
RESPONSE: We thank the reviewer for this helpful feedback. We have now added an additional paragraph (see below and on page 9).
RAGE activation after TBI contributes to neurovascular unit damage, BBB disruption, and neuroinflammation. Inhibiting RAGE in animal models, via gene knockouts genetically or with antagonists like FPS-ZM and papaverine, reduces BBB damage, brain edema, and neuroinflammation. In cellular models, RAGE is upregulated in brain tissue after TBI, especially in microglia and pericytes near the injury site. Blocking RAGE reduces the number of activated glial cells and apoptotic cells, while increasing neuron survival. Multiple miRNAs, including miR-214, miR-185-5p, miR-107, miR-21, miR-155, and miR-181a, are known to modulate RAGE or its signaling pathways in cancer biology, while their role in neuroinflammation or TBI remains unknown.
7. In this regard, targeting not S100b itself, but its targets, could be a more effective approach. The authors should discuss this in the text and justify the appropriateness of the approach they propose within the framework of the review.
RESPONSE: We thank the reviewer for this helpful feedback. We have now added an additional paragraph (see below and on page 9) that justifies our focus on targeting S100B using microRNA.
Here, we primarily focus on inhibition of S100β mRNA translation through the employment of S100β-targeting miRNAs due to three factors: (1) the complexity of RAGE-dependent signaling makes it a tricky therapeutic target[92,100], (2) S100B levels are greatly elevated after a TBI, and (3) S100B is neurotrophic at lower concentrations and neurotoxic depending at higher concentrations, thereby modulation of this protein would be a good therapeutic goal.
Round 2
Reviewer 3 Report
Comments and Suggestions for Authors
The authors have addressed all of my comments. However, the added fragment about role of S100b in microglia functioning has to be slightly corrected:
"As the resident immune cell of the brain". "cells" is a correct variant.
-"The M1 phenotype is activated by signals from interferon gamma (IFN-y), TNF-a, and IPS to lead...". Maybe LPS instead of IPS?
"M2 microglia are associated with anti-inflammatory and pro-survival cytokines in response to interleukin-4 (IL-4) and interleukin-13 (IL-13)". Incomplete sentence.
"Microglia in the M2 state clear out cell debris and resolve damage.". Incorrect phrases "clear out cell debris" and "resolve damage".
"encourage a prolonged M microglia". Which type is meant?
"increased infarct size, TUNEL-positive neurons,". Better "the percentage of TUNEL-positive neurons".
"Modulating S100β levels could prove to be a beneficial mechanism of modulating the balance between M1 and M2 microglia populations in various neuropathies". Please, revise this sentence.
Author Response
"As the resident immune cell of the brain". "cells" is a correct variant.
RESPONSE: Thank you for this correction. It has now been corrected.
-"The M1 phenotype is activated by signals from interferon gamma (IFN-y), TNF-a, and IPS to lead...". Maybe LPS instead of IPS?
RESPONSE: Thank you for noticing this typo. It has now been updated.
"M2 microglia are associated with anti-inflammatory and pro-survival cytokines in response to interleukin-4 (IL-4) and interleukin-13 (IL-13)". Incomplete sentence.
RESPONSE: Thank you for this correction. It has now been updated as, ".... associated with the release of anti-inflammatory..."
"Microglia in the M2 state clear out cell debris and resolve damage.". Incorrect phrases "clear out cell debris" and "resolve damage".
RESPONSE: Thank you for this correction. It has now been updated as, "Microglia in the M2 state help repair tissue by clearing cellular debris and modulating neuroinflammation."
"encourage a prolonged M microglia". Which type is meant?
RESPONSE: Thank you for this correction. It has now been updated as, "However, the accumulation of misfolded or aggregated proteins promotes a sustained pro-inflammatory M1 microglial state, which worsens inflammation and contributes to neuronal cell death"
"increased infarct size, TUNEL-positive neurons,". Better "the percentage of TUNEL-positive neurons".
RESPONSE: Thank you for this correction. It has now been updated.
"Modulating S100β levels could prove to be a beneficial mechanism of modulating the balance between M1 and M2 microglia populations in various neuropathies". Please, revise this sentence.
RESPONSE: Thank you for this correction. It has now been updated as, "Modulating S100β levels may help shift the balance between M1 and M2 microglial states, offering a potential therapeutic strategy after brain injury or ischemia. "
